# Comparison of the Clinical Results of Platelet-Rich Plasma, Steroid and Autologous Blood Injections in the Treatment of Chronic Lateral Epicondylitis

**DOI:** 10.3390/healthcare11050767

**Published:** 2023-03-06

**Authors:** Aybars Kıvrak, Ibrahim Ulusoy

**Affiliations:** 1Adana Avrupa Hospital, Adana 01170, Turkey; 2Selahhadin Eyyubi State Hospital, Diyarbakır 21100, Turkey

**Keywords:** chronic, lateral epicondylitis, steroid, PRP, autologous whole blood injection, injection, elbow, VAS

## Abstract

Background: The aim of our study is to compare the efficacy of PRP, steroids, and autologous blood injection in patients with chronic lateral epicondylitis. Method: A total of 120 patients comprised our study. Three groups of 40 patients each received only one of the following: PRP, steroids, or autologous blood injections. Thereafter, VAS (visual analog scale), DASH (Disabilities of the Arm, Shoulder, and Hand), and Nirschl scores of those treated were evaluated in the second week, the fourth week, the third month, and the sixth month. Results: The baseline evaluation revealed no significant change in VAS, DASH, and Nirschl scores among the three groups (*p* > 0.050). At the second week evaluation, patients treated with steroids showed significant improvement compared to patients treated with PRP and autologous blood (*p* < 0.001). The fourth-week evaluation revealed the VAS, DASH, and Nirschl scores of the patients treated with steroids to have improved more significantly than those of patients treated with PRP and autologous blood (*p* < 0.001). The third month, when the results of all three groups were compared, revealed similar results (*p* > 0.050). The sixth-month evaluation, when the results of all three groups were compared, revealed the autologous blood and PRP applications provided significantly better results than the group treated with steroids (*p* < 0.001). Conclusion: We concluded that steroid administration is effective in the short term, while PRP and autologous blood applications are more effective than steroid administration in the long term.

## 1. Introduction

Lateral epicondylitis, also known as tennis elbow, is a clinical condition seen in approximately 1–3% of the population. The disease usually affects individuals between the ages of 35 and 50 [1,2]. The underlying cause of this pathology is often not fully revealed. Overuse of the wrist extensor and supinator muscles is among the most common causes, although it is frequently seen in tennis players. The most affected muscle is extensor carpi radialis brevis. The diagnosis of lateral epicondylitis is usually made by physical examination. Pain with pressing on the lateral epicondylitis in the elbow and a positive Cozen’s test occupies an important place in the diagnosis. Also, resisted wrist extension with the elbow fully extended can exacerbate pain. Rather than a simple inflammatory event, lateral epicondylitis is a complex pathology in the lateral epicondyle region of the humerus at the attachment site of the extensor tendons (extensor carpi radialis brevis, extensor digitorum, extensor digiti minimi, and extensor carpi ulnaris) in the forearm [3].

Many different treatment methods have been described in the treatment of lateral epicondylitis. Activity modification, NSAID use, bracing, extracorporeal shock wave therapy, YAG laser treatment, and acupuncture are among the treatment options. Manual therapy, TECAR therapy, ultrasound and TENS applications, and exercises (passive, active-assisted, active, stretching, etc.) can also be counted. Botulinum toxin, PRP, steroids, and autologous whole-blood injections are also frequently used in lateral epicondylitis [4]. Despite the wide range of treatment options, there is still no consensus in the literature as to the most efficacious treatments, especially in chronic cases [5,6,7]. Despite the varied results published in the literature relating to PRP, steroids, and autologous whole-blood injections, no published study has compared the efficacy and safety of these three treatment methods.

## 2. Methods

Our retrospective study scanned our hospital information management system to review cases originating between 2017 and 2020. Patients evidencing pain with pressure to the lateral epicondyle; pain with active wrist extension against resistance in forearm pronation; pain with full elbow extension; and a diagnosis of lateral epicondylitis were evaluated for inclusion in the study. Patients admitted to our hospital with a diagnosis of lateral epicondylitis whose complaints did not improve despite traditional treatments (NSAIDs, physical therapy, bracing, etc.) and who had undergone elbow injections for lateral epicondylitis were included in the study, as were patients who were examined, followed up, and treated by a single clinician. Where files contained incomplete or suspicious data, patients were telephoned to confirm their occupation, dominant arm, and affected arm.

Criteria for inclusion in the study included patients who were followed up for three months with traditional treatment methods (such as NSAID use, physical therapy applications, use of brace); patients whose complaints did not resolve after treatment; patients who were 17 years of age or older and had direct radiographs and no pathologies (e.g., intra-articular free body, osteoarthrosis); and patients who attended the polyclinic controls on the fifteenth day, fourth week, third month, and sixth month and for whom no other treatment occurred during these follow-ups. Criteria for exclusion from the study included patients with a history of acute elbow trauma; diagnosis of systemic diseases such as rheumatoid arthritis, DM (diabetes mellitus), hepatitis, anemia, and bleeding disorders; patients who must use anticoagulant or antiaggregant treatment; patients with severe weight change in the last year (i.e., size change in the patient’s daily life clothes); patients with a history of malignancy, pregnancy, previous elbow fractures, epilepsy, or cervical radiculopathy; patients with a history of elbow surgery; patients with an elbow injection history in the last six months; patients with additional pathologies in the elbow (such as pin neuropathy); patients with diagnosed psychiatric disorders; patients lacking necessary data in the patient file for the study; and patients unable to communicate.

The minimum number of patients to be included in the study groups was determined through power analysis. Afterward, the patients were divided into subgroups by making stratified sampling according to gender from the determined patient group pools.

With the patients’ elbows flexed at 90 degrees, 2 mL autologous whole blood + 1 mL prilocaine hydrochloride (20 mg/mL), 2 mL prilocaine hydrochloride (20 mg/mL) + 1 mL betamethasone (injection contains betamethasone dipropionate equivalent to 5 mg betamethasone and betamethasone sodium phosphate equivalent to 2 mg betamethasone), or 2 mL PRP + 1 mL prilocaine hydrochloride (20 mg/mL) were applied with the help of 21 G injector to the area with the most pain on the lateral epicondyle using the technique of peppering. Afterward, patients in all groups were advised to rest their elbows the first day after injection, to adhere to immobilization with a shoulder arm sling, to avoid anti-inflammatory drugs other than paracetamol the first two weeks after injection, to avoid cold or hot application, and to avoid anticoagulant and anti-aggregate drugs during the follow-up. The patients were warned during the follow-up against making additional injections to the elbow area. Control polyclinic data of the patients were evaluated on the fifteenth day, fourth week, third month, and sixth month. VAS, DASH, Nirschl (Numeric Pain Intensity Scale) scores, and clinical results were evaluated in the pre- and post-injection follow-ups.

### 2.1. PRP and Whole Blood Preparation

Preparation and administration of platelet-rich plasma were performed in all patients under the same conditions, according to the method described by Anitua et al. [8,9]. A total of 30 mL of peripheral blood was taken from the antecubital region. The sample was placed in tubes containing 3.2% sodium citrate and taken into another tube with EDTA to check the number of peripheral platelets. Samples were centrifuged at 1500 rpm for 8 min at room temperature. The amount of 1 mL of the PRP obtained was sent to the laboratory for platelet count. Activated PRP (50 μL Cl_2_ Ca for 1 mL PRP) was applied by palpation under sterile conditions to the most sensitive point in the lateral epicondyle area at the elbow.

For autologous blood injection, 2 mL of venous blood taken from the other upper extremity of the patient was mixed with 1 mL of prilocaine hydrochloride (20 mg/mL) and made ready for injection.

### 2.2. Statistical Analysis

Statistical analysis was performed using SPSS software (IBM Corp. Released 2017. IBM SPSS Statistics for Windows, Version 25.0. Armonk, NY, USA: IBM Corp.). The conformity of the numerical variables to the normal distribution was performed using visual (histogram and probability graphs) and analytical methods (Kolmogorov-Smirnov/Shapiro-Wilk tests). The descriptive statistics for numerical variables showing normal distribution in the comparisons were given by mean and standard deviation. Descriptive statistics of categorical variables were given by numbers and percentage values. One-way ANOVA was used to compare the parameters, and Paired Student’s *t*-test was used to compare the measurements at two-time points. The Statistical significance level was accepted as *p* < 0.001.

Clinical research ethics committee approval was obtained for this study. The ethics committee number is E-10879717-050.01.04-2580.

## 3. Results

Considering the inclusion and exclusion criteria, the data of 147 qualifying patients were obtained. Forty patients, 20 male and 20 female, were selected for each of three groups by stratified sampling according to gender. A total of 120 patients were included in the study. When the mean age, female-male distribution of PRP, steroid, and autologous blood injection groups, dominant sides of the individuals, and the affected extremity were compared, no significant difference was found (*p* > 0.050) (Table 1) (Figure 1).

Analysis of the samples of all the patients undergoing PRP showed that the platelet count of the applied PRP content was at least 2.5 times higher than the number of platelets in the peripheral blood, and the platelet count of the applied PRP content was above 300,000 platelets/μL [9,10].

The pre-injection VAS score, PRTEE, and DASH scores of the three groups that became the subjects of this study are similar, and no statistically significant difference was found between them (*p* > 0.050) (Table 2).

In the statistical evaluation, when the VAS, DASH, and Nirschl scores of the patients who underwent PRP were compared on day zero and day 15, no significant change was found (*p* > 0.050). Day zero and day 15 comparison of the VAS, DASH, and Nirschl scores of patients undergoing autologous blood injection showed no significant change (*p* > 0.050). Day zero and day 15 comparison of the VAS, DASH and Nirschl scores of the patients undergoing steroid application showed a significant decrease (*p* < 0.001). Day 15 comparison of the VAS, DASH, and Nirschl scores of the patients treated with steroids, as well as the patients treated with PRP and autologous blood injection, showed a significant improvement (*p* < 0.001).

When Day 15 and Week Four values were compared, a significant decrease in the DASH score, a significant decrease in the VAS score, and a significant decrease in the Nirschl score were found in patients who underwent PRP (*p* < 0.001). In patients who underwent autologous blood injection, a significant decrease in DASH score (*p* < 0.001), a significant decrease in VAS score (*p* < 0.001), and the same Nirschl score (*p* > 0.050) were found. In patients treated with steroids, a significant decrease in DASH score (*p* < 0.001), a significant decrease in VAS score (*p* < 0.001), and a significant decrease in the Nirschl score (*p* < 0.001) were found. At the fourth week evaluation, patients who received steroids showed a significantly greater improvement in VAS, DASH, and Nirschl scores compared to the scores of the patients undergoing PRP and autologous blood injections (*p* < 0.001).

When the scores of the fourth week and the third month were compared, a significant decrease in the DASH score (*p* < 0.001), a significant decrease in the VAS score (*p* < 0.001), and a significant decrease in the Nirschl score (*p* < 0.001) were found in patients who underwent PRP. In patients undergoing autologous blood injection, a significant decrease in DASH score (*p* < 0.001), a significant decrease in the VAS score (*p* < 0.001), and a significant decrease in the Nirschl score (*p* < 0.001) were found. In patients treated with steroids, there was a significant increase in the VAS score (*p* < 0.001) and a significant increase in the Nirschl score (*p* < 0.001), but the DASH score was the same (*p* > 0.050). Similar results were obtained when the results of all three groups were compared (*p* > 0.050).

A comparison of the values of the third month and the sixth month showed a significant decrease in the DASH score (*p* < 0.001), a significant decrease in the VAS score (*p* < 0.001), and a significant decrease in the Nirschl score (*p* < 0.001) in patients who underwent PRP. A significant decrease in the DASH score (*p* < 0.001), a significant decrease in the VAS score (*p* < 0.001), and a significant decrease in the Nirschl score (*p* < 0.001) were seen in patients who underwent autologous blood injections. A significant increase in the DASH score (*p* < 0.001), a significant increase in the VAS score (*p* < 0.001), and the same values in the Nirschl score (*p* > 0.05) were seen in patients treated with steroids. A comparison of the results of the three groups at the sixth-month evaluation determined that the autologous blood and PRP administrations resulted in significantly better outcomes than the steroid-applied group (*p* < 0.001). No local or systemic complications were observed in any of the patients during the application and follow-up.

## 4. Discussion

Our study examined the VAS, DASH, and Nirschl scores of patients diagnosed with lateral epicondylitis who received autologous blood, PRP, and steroid injections to their elbow area and who were followed up on the fifteenth day, in the fourth week, third month, and sixth month after injection. According to the results of our study, although there was a significant early regression in the complaints of patients treated with steroids, the severity of complaints increased in the fourth week, and the complaints recurred during the six-month follow-up evaluation. In contrast, there was no early improvement in complaints with PRP and autologous blood injection, and gradual improvement was observed in patients’ complaints after the fourth week (Figure 2, Figure 3 and Figure 4).

In the treatment of lateral epicondylitis, activity modification, ice application, and non-steroid medication can be used. There are also treatment methods such as steroid, PRP, and autologous blood injection to the elbow area. Though previously considered to be the gold standard of treatment, steroid injection is now controversial [4]. In a randomized controlled study involving 164 patients, the treatment of lateral epicondylitis with local corticosteroid injections, standard NSAID drugs, and simple analgesics was examined [11]. Better results were obtained in patients receiving steroids compared with the other two groups in the fourth week of the study. In the evaluation made in the twelfth month, similar results were obtained in three groups, and it was concluded that the steroid was effective in the early period but did not have an advantage in the long term. In another randomized controlled study involving 185 patients, patients were divided into three groups: corticosteroid injection, physiotherapy, and wait-and-see policy were applied in their treatment [12]. In the sixth week, better results were obtained in the steroid group than in the other groups. However, in follow-up over 52 weeks, 69% success was achieved for steroid injection, 91% for physiotherapy, and 83% for wait-and-see policy. The long-term results led to the conclusion that steroid injection was unsuccessful compared to physiotherapy and wait-and-see policy. In another study in which 34 people participated, patients were randomized to three groups: one in which splinting and stretching were applied, another in which cortisone injection was administered, and a third in which deep friction massage was administered [13]. The study observed improvement in the VAS pain score in all: in the DASH and grip strength scores in the cortisone injection group; and in the deep friction massage group during short-term follow-ups (6–12 weeks). In the six-month follow-up, a significant improvement was observed only in the deep friction massage group. Although the use of steroids is a frequently preferred method in the treatment of lateral epicondylitis, it has been observed that it is effective in the early period of 4–6 weeks in the light of the current literature, but it is not effective in regressing pain and improving functions in the long-term [1,7,11,12,13,14,15]. It is thought that the effective factor in reducing pain in the early period of the steroid use is the suppression of inflammation with the media of the arachidonic acid pathway [16]. In reference to the long-term results, an experimental study on rats suggested additionally that long-term steroid therapy causes a progressive thinning of the peroneus longus tendon and a significant decrease in the amount of collagen in the tendon, due to inhibition of collagen synthesis [17]. Our study correlates with the literature in finding that steroids are effective in reducing pain and improving functions in the short term but do not have the same effectiveness in the long term.

Recent histological findings have shown that tendinosis is not an acute inflammatory condition but a pathology in which normal tendon repair fails. Researchers have thus turned to biological treatment methods such as autologous whole blood injection and PRP application [1]. Various studies have demonstrated the efficacy of these treatments. In a study involving 21 men and 59 women, the results of patients diagnosed with lateral epicondylitis were evaluated after autologous blood and corticosteroid injection [18]. After the fifteenth, thirtieth, and ninetieth day follow-ups, patients’ elbow pain was evaluated in the sixth month using VAS scores established in telephone interviews. These results led to the conclusion that autologous blood injections reduce pain and improve functions and grip strength more effectively than corticosteroid injections. Another study divided 60 patients into three groups and treated them with corticosteroid injections, autologous blood injections, and extracorporeal shock wave therapy [19]. A high rate of success was achieved with corticosteroid injections in the short term, but high recurrence rates were observed in the long term. In addition, better outcomes resulted from autologous blood injections and extracorpereal shock wave therapy. Autologous blood injections were recommended in the treatment of lateral epicondylitis. Another study randomized 70 lateral epicondylitis patients to control and injection groups and shared the first, third, and sixth month follow-up results, concluding that the benefits of autologous blood injections to such patients continue in the sixth month and, therefore, may be considered among the long-term treatment options for lateral epicondylitis [20]. A study in which 31 patients were followed up for an average of 5.2 years included patients who did not benefit from conservative treatment [21]. This study concluded that a single dose PRP injection was effective in the treatment of many patients diagnosed with refractory lateral epicondylitis and reduced the need for surgery. A double-blind prospective randomized controlled study divided 230 patients into two groups, a PRP group and the active control group. The 12-week follow-ups showed no difference between the study groups, but the 24-week follow-ups showed clinically significant improvement in the PRP group [22]. Another study divided 150 patients into three groups and administered corticosteroids to one group, xylocaine to a second group, and PRP to a third group [14]. Patients were followed up with VAS and Nirschl staging at week zero, two, six, 12, 26 and 52. The study found PRP, corticosteroid, and xylocaine to be safe and effective in the treatment of lateral epicondylitis. It was concluded that steroids and xylocaine were effective in the short term, but PRP was more effective and permanent in relieving pain than the other two treatment methods in long-term follow-up. In addition to these studies, a randomized double-blind controlled study divided 60 patients into three groups, each receiving different treatments: PRP, saline, and glucocorticoid injections [5]. According to the results, no superiority of PRP and glucocorticoid to saline injection was found in terms of pain reduction in the three-month follow-up. However, after one month of follow-up, glucocorticoid therapy was found to be superior to other treatment options for pain reduction. In another study, 119 patients were divided into three groups, and three different treatments were applied: PRP, saline, and autologous whole blood injection [6]. In the one-year follow-up, no significant improvement was found in the pain and function levels of the patients when the groups with autologous blood and PRP injections were compared with the group that received saline injection.

Unlike corticosteroids, PRP and autologous blood injections aim to trigger inflammation rather than suppress it [23,24,25]. PRP and autologous blood contain platelets. The platelets contain powerful growth factors and granules that are effective in chronic injuries [23,24,25]. These growth factors trigger stem cell replenishment, increase local vascularization, and stimulate collagen production by tendon sheath fibroblasts; concordantly, an increase in endogenous growth factor production was observed in human tendons treated with PRP [26,27,28]. This mechanism explains that a single dose of PRP or autologous blood administration has a permanent effect on the healing process. Our study demonstrated that, in accordance with many publications in the literature, PRP and autologous blood injection are effective treatments for reducing pain and improving functions in long-term follow-up but are poor treatments in the early period compared to steroid injection. Although we find long-term results of PRP and autologous blood injection treatments statistically similar, autologous blood injection might be recommended, considering the small amount of sampling (2ml VS 30 mL), no cost of processing, and equal long-term effect. There is no consensus in the literature about the role of biological methods such as PRP and autologous blood injection in the treatment of lateral epicondylitis [5,6,14,18,19,20,22]. Factors such as the number and frequency of injections, the volume of injections given to the patient, the method and placement of the injection, method of preparation and the cell density of PRP, post-injection protocols, patient’s nutritional habits, and ethnicity of the patient might change the results of these studies. Further studies, thus, are needed.

The limiting factors of our study are the lack of histopathological and radiological examinations; the use of data belonging to a single center and to people living in a single region; the lack of a standard treatment protocol surrounding dose, frequency, and number of PRP injection applications in the literature; and the limitation of patient follow-up past six months.

## 5. Conclusions

Although PRP and autologous blood injections are not as effective as steroid injections in the short term, they are more effective in the long-term treatment of lateral epicondylitis. Performing these treatments with autologous biological agents increases the trust people place in these treatments, but the low density of platelets in autologous blood compared to PRP decreases its effectiveness; the use of special devices and the time and costs required to prepare PRP can also be counted among its disadvantages. In order to obtain a standard treatment protocol for lateral epicondylitis, studies with longer follow-up periods and clinical, radiological, and histopathological evaluations are needed in large patient groups.

## Figures and Tables

**Figure 1 healthcare-11-00767-f001:**
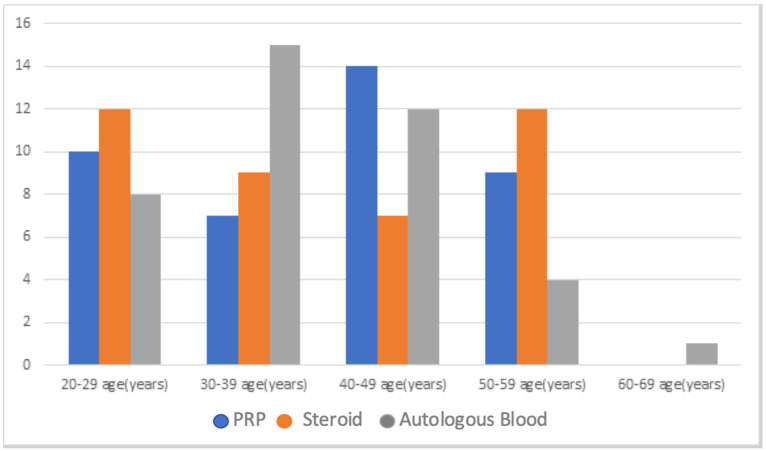
Distribution of patient groups by age.

**Figure 2 healthcare-11-00767-f002:**
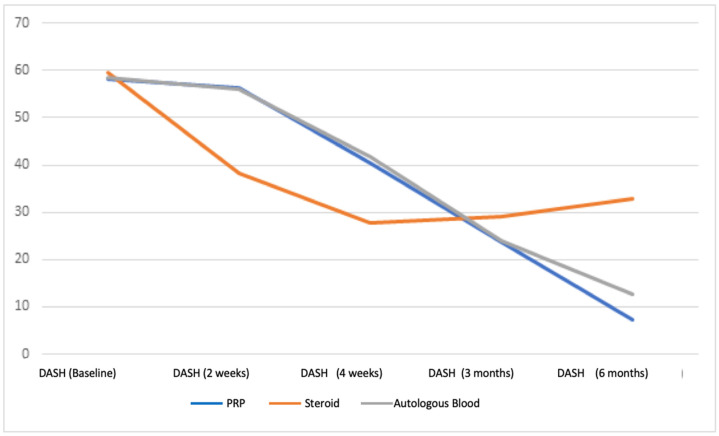
Mean unit improvement in DASH score.

**Figure 3 healthcare-11-00767-f003:**
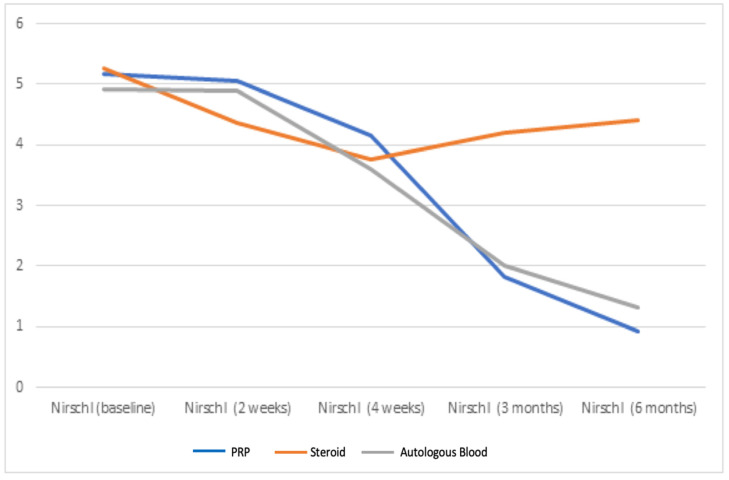
Mean unit improvement in Nirschl score.

**Figure 4 healthcare-11-00767-f004:**
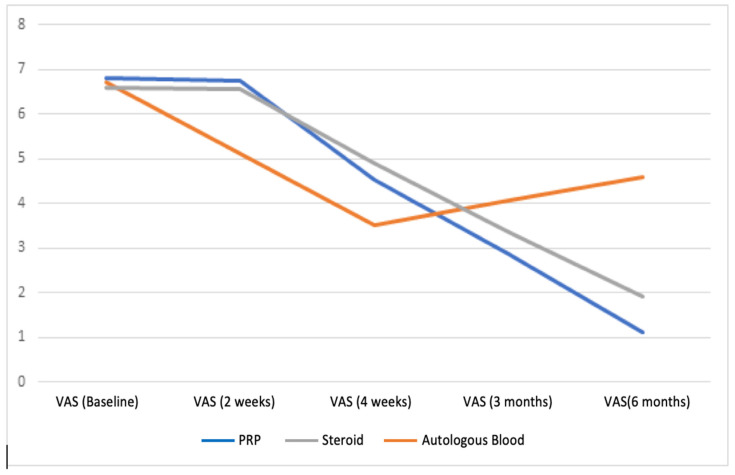
Mean unit improvement in visual analog scale pain score.

**Table 1 healthcare-11-00767-t001:** Distribution of patients by age, gender, affected arm, and dominant arm.

	PRP	Steroid	Autologous Blood	*p* Value
Age (years, mean)	40.13 ± 10.38	39.67 ± 11.19	38.65 ± 8.96	*p* = 0.804
Gender (male/female)	20/20	20/20	20/20	*p* = 1
Side (Right/Left)	31/9	35/5	35/5	*p* = 0.371
Dominant arm (right/left)	36/4	40/0	37/3	*p* = 0.141

**Table 2 healthcare-11-00767-t002:** VAS, DASH, and Nirschl score changes of patients over time.

	PRP	Steroid	Autologous Blood	*p* Value
VAS				
Baseline	6.8 ± 1.03	6.7 ± 0.82	6.6 ± 0.87	*p* = 0.546
2 weeks	6.75 ± 1.39	5.12 ± 0.85	6.57 ± 1.21	*p* < 0.001
4 weeks	4.52 ± 0.93	3.5 ±0.93	4.95 ± 1.51	*p* < 0.001
3 months	2.87 ± 0.75	4.05 ± 0.87	3.37 ± 0.92	*p* < 0.001
6 months	1.12 ± 1.09	4.6 ± 1.1	1.09 ± 1.46	*p* < 0.001
DASH				
Baseline	58.16 ± 6.13	59.67 ± 7.73	58.54 ± 5.92	*p* = 0.575
2 weeks	56.2 ± 8.03	38.26 ± 10.96	56.17 ± 9.16	*p* < 0.001
4 weeks	40.33 ± 7.23	27.85 ± 8.83	41.68 ± 8.48	*p* < 0.001
3 months	23.67 ± 4.24	29.13 ± 7.67	24 ± 4.57	*p* < 0.001
6 months	7.21 ± 8.19	32.84 ± 8.99	12.75 ± 7.5	*p* < 0.001
NIRSCHL				
Baseline	5.1 ± 0.90	5.27 ± 0.59	4.92 ± 0.88	*p* = 0.142
2 weeks	5.05 ± 0.9	4.37 ± 1	4.9 ± 0.95	*p* = 0.005
4 weeks	4.15 ± 0.89	3.7 ± 0.89	3.6 ± 1.15	*p* = 0.041
3 months	1.82 ± 0.87	4.2 ± 0.72	2 ± 0.84	*p* < 0.001
6 months	0.92 ± 1.07	4.42 ± 1.08	1.32 ± 1.07	*p* < 0.001

## Data Availability

The data presented in this study are available on request from the corresponding author. The data are not publicly available due to using these study data, a different study can be done in a larger patient group. For detailed information, the corresponding author can be contacted.

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
