# Peer review of "Comparison of the Clinical Results of Platelet-Rich Plasma, Steroid and Autologous Blood Injections in the Treatment of Chronic Lateral Epicondylitis"

_healthcare, 2023, doi:10.3390/healthcare11050767_

Round 1

Reviewer 1 Report

This is a good manuscript comparing PRP, steroid, and autologous blood. 

# Line 116~121 and Table 1. 

In this retrospective study, 147 patients were included. I would like to ask the authors how to assign the very similar number of patients to each group.  

# In the result, the authors did not suggest subgroup analysis. In one-way ANOVA, a significant p-value shows that at least one group has a different result, but cannot show one is better than the others. Please perform subgroup analysis.

# In line 261. RPR ? or PRP?

# In lines 281~284, the platelet count is not a factor to check effectiveness. I think this sentence is not appropriate.  On the contrary, according to this result, I think whole blood injection might be recommended considering the small amount of sampling (2ml VS 30 ml), no cost of processing, and equal long-term effect. 

Author Response

Point 1-Line 116~121 and Table 1. 

In this retrospective study, 147 patients were included. I would like to ask the authors how to assign the very similar number of patients to each group.  

Response 1-First of all, power analysis was performed to determine the minimum number of patients that should be included in the groups. After a 3-year archive scan, the patients were divided into subgroups by making stratified sampling according to gender from the determined patient group pools (Line75-76). If the minimum number of patients in each group could not be achieved, a 4- or 5-year retrospective review would be performed.

Point 2-# In the result, the authors did not suggest subgroup analysis. In one-way ANOVA, a significant p-value shows that at least one group has a different result, but cannot show one is better than the others. Please perform subgroup analysis.

response 2-

As you mentioned, if there is a significant difference in the one-way Anova test, the post-hoc tests to be selected according to the Levene test result (tests of homogeneity of variances) will determine which group the significant difference originates from. The reason for not going into detail in the results section is to prevent it from being boring for the readers. In our study, the results were obtained by performing post-hoc tests.

Point-3 # In line 261. RPR ? or PRP?

response 3- PRP. corrected in the document.

Point-4 # In lines 281~284, the platelet count is not a factor to check effectiveness. I think this sentence is not appropriate.  On the contrary, according to this result, I think whole blood injection might be recommended considering the small amount of sampling (2ml VS 30 ml), no cost of processing, and equal long-term effect. 

Response-4.We removed the sentence you mentioned. Edited according to your suggestions.

"Although we find long-term results of PRP and whole blood injection treatments statistically similar, whole blood injection might be recommended considering the small amount of sampling (2ml VS 30 ml), no cost of processing, and equal long-term effect."

Author Response

Point-1 Comment: Abstract and throughtout manuscript: Eith use "autologous blood" or "whole blood" decide which one to use

Reponse-1. Corrections were made and autologous blood is used in the text.

Point-2 Comment: Mention any technique used for whole blood

Response-2.

"For autologous blood injection, 2 ml of venous blood taken from the other upper extremity of the patient was mixed with 1 ml of prilocaine hydrochloride (20mg/ml),  and made ready for injection."

Added to the text.

Point-3.line 303.change "long" to "longer"

response-3.Corrections were made

Reviewer 3 Report

Thanks to the author for their effort in writing this paper.

The lateral epicondylitis is a very common pathology with an important prevalence, so the topic is of great interest.

The paper is well written, with a good architecture and very clear, straightforward to the point. However I'd advise minor revisions.

I'd like to read something about also physical therapy (i.e. Laser Yag, manipulations, exercises, TECAR treatments) that can be effective in this pathology, and was never cited by the authors. That could add a more complete overview of the treatment options for this pathology.

Moreover, no explanation of the etiology and diagnosis of the pathology was given. I'd add some sentences on the introduction section.

The points of strenght of this paper are that the pathology addressed in the paper is very common, so it is a topic of great interest. Moreover, the statistical analysis is well described and supports the conclusion stated by the authors. The points of weakness are that the authors must describe better the pathology, describing its diagnosis and etiology, and describe better the therapeutic options for this pathology including the physical therapy.

Thank you 

Author Response

point 1:

I'd like to read something about also physical therapy (i.e. Laser Yag, manipulations, exercises, TECAR treatments) that can be effective in this pathology, and was never cited by the authors. That could add a more complete overview of the treatment options for this pathology.

Moreover, no explanation of the etiology and diagnosis of the pathology was given. I'd add some sentences on the introduction section.

response-1:With your suggestions, necessary additions were made in the introduction part.

Round 2

Reviewer 1 Report

corrected well.